# Coding and Non-Coding RNA Abnormalities in Bipolar Disorder

**DOI:** 10.3390/genes10110946

**Published:** 2019-11-19

**Authors:** Jurjen J. Luykx, Fabrizio Giuliani, Giuliano Giuliani, Jan Veldink

**Affiliations:** 1Department of Translational Neuroscience, Brain Center Rudolf Magnus, University Medical Center Utrecht, Utrecht University, 3584 CX Utrecht, The Netherlands; f.giuliani30@gmail.com (F.G.); giuliano.giuliani83@gmail.com (G.G.); j.h.veldink@umcutrecht.nl (J.V.); 2Department of Psychiatry, Brain Center Rudolf Magnus, University Medical Center Utrecht (UMCU), Utrecht University, 3584 CX Utrecht, The Netherlands; 3GGNet Mental Health, 7328 JE Apeldoorn, The Netherlands; 4Department of Neurology, Brain Center Rudolf Magnus, University Medical Center Utrecht (UMCU), Utrecht University, 3584 CX Utrecht, The Netherlands

**Keywords:** bipolar disorder, circular RNA, long non-coding RNA, alternative splicing, cEPHA3, histone H3-K4 demethylation, sequencing

## Abstract

The molecular mechanisms underlying bipolar disorder (BPD) have remained largely unknown. Postmortem brain tissue studies comparing BPD patients with healthy controls have produced a heterogeneous array of potentially implicated protein-coding RNAs. We hypothesized that dysregulation of not only coding, but multiple classes of RNA (coding RNA, long non-coding (lnc) RNA, circular (circ) RNA, and/or alternative splicing) underlie the pathogenesis of BPD. Using non-polyadenylated libraries we performed RNA sequencing in postmortem human medial frontal gyrus tissue from BPD patients and healthy controls. Twenty genes, some of which not previously implicated in BPD, were differentially expressed (DE). PCR validation and replication confirmed the implication of these DE genes. Functional in silico analyses identified enrichment of angiogenesis, vascular system development and histone H3-K4 demethylation. In addition, ten lncRNA transcripts were differentially expressed. Furthermore, an overall increased number of alternative splicing events in BPD was detected, as well as an increase in the number of genes carrying alternative splicing events. Finally, a large reservoir of circRNAs populating brain tissue not affected by BPD is described, while in BPD altered levels of two circular transcripts, cNEBL and cEPHA3, are reported. cEPHA3, hitherto unlinked to BPD, is implicated in developmental processes in the central nervous system. Although we did not perform replication analyses of non-coding RNA findings, our findings hint that RNA dysregulation in BPD is not limited to coding regions, opening avenues for future pharmacological investigations and biomarker research.

## 1. Introduction

Bipolar disorder (BPD) is characterized by one or more manic episodes alternating with euthymia or depressive episodes. Bipolar spectrum disorders carry a lifetime prevalence of up to 2% and are estimated to be the seventeenth leading cause of disability worldwide [1]. They entail increased risks of comorbid conditions and premature death (in part owing to suicide) [2,3]. The diagnosis of BPD is often challenging due to its clinical heterogeneity and, consequently, diagnostic delays are common. Moreover, response to currently available treatment modalities is seldom complete and burdensome adverse reactions often emerge. Such diagnostic pitfalls and imperfect treatments may all benefit from neurobiological studies aimed at increasing our understanding of its pathophysiology.

To date, the biological mechanisms underlying BPD remain largely elusive, although research hints that both genetic and environmental factors contribute. For example, twin-based heritability estimates vary between 70 and 90% [4]. Recent studies aimed at elucidating the molecular basis of BPD have mostly relied on genetic approaches, including genome-wide association (GWA) [5,6,7] and whole-exome sequencing studies [8,9,10], to identify both inherited and *de novo* variation contributing to BPD. The best powered GWAS has highlighted 30 loci for BPD and has provided insight into genes and pathways involved in the disease [7]. Therefore, gene expression analysis of the relevant brain regions constitutes a primordial step to help identify the molecular pathways altered in BPD.

In one of the first comprehensive gene expression analyses in BPD, peripheral blood cells for microarray-based transcriptome analysis were used to identify changes in levels of transcripts involved in G-protein signalling [11]. More recently, next generation sequencing (NGS) technologies have been used to survey the brain transcriptome in bipolar disorder, in particular by the PsychENCODE consortium (http://resource.psychencode.org/), with intriguing results for a range of psychiatric disorders (https://science.sciencemag.org/content/362/6420) [12]. RNA sequencing of hippocampus, the anterior cingulate gyrus, the dorsolateral prefrontal cortex, and the dorsal striatum of BPD postmortem tissue has moreover identified genes linked to G-protein coupled receptors, circadian rhythm, the immune system, inflammatory response and metabolic pathways [13,14,15,16,17,18]. However, most of these RNAseq experiments were designed to capture polyadenylated RNA transcripts—which include protein-coding mRNAs and a number of non-coding RNAs -, while most RNAs (>90% of the transcriptome) do not carry a polyadenylated tail. On a similar note, NGS experiments indicate that less than 5% of transcription across the human genome results in protein-coding genes, while the remaining pool is associated with non-protein coding transcripts [19], approximately 60% of which belong to the class of long non-coding RNAs (lncRNAs) [20]. To date, only a few lncRNAs have been characterized at the molecular or functional level but their dysregulation is being increasingly reported in cancer and in numerous neurological, cardiovascular, and developmental diseases [21,22,23,24,25]. Furthermore, although the dorsolateral prefrontal gyrus has been targeted [13], other sections of the (pre)frontal gyrus have been left unused in RNA sequencing studies of postmortem brain tissue in BPD patients. This absence of frontal gyrus RNA sequencing studies in BPD is in sharp contrast to the currently available impressive body of literature hinting at the implication of particularly the medial frontal cortex in BPD. For example, meta-analytic evidence points to medial frontal gray matter reductions in BPD compared to controls [26], resting-state connectivity aberrations in the medial frontal cortex [27], and altered activity in this area in BPD based on fMRI studies [28]. Thus, because the medial part of the frontal gyrus has been particularly implicated in BPD by a range of studies we set out to obtain frozen sections of this brain region for RNA sequencing.

Sequencing experiments of non-polyadenylated transcriptomes have led to the discovery of new RNA classes, such as circular RNAs (circRNAs), a category of lncRNAs produced by back-splicing reactions that covalently link the 3′ end of an exon to the 5′ end of an upstream exon [29,30,31]. circRNAs have been implicated in gene regulation, by functioning as molecular sponges to regulate gene expression of microRNAs, sequestering RNA binding proteins and competing with other lncRNAs [32,33,34]. Recent studies have shown that circRNAs and other lncRNAs also play pivotal roles in brain development and neuronal integrity [35,36,37,38,39,40,41]. Non-polyadenylated RNAseq libraries also allow probing of alternative splicing, a process that not only generates protein diversity, but also constitutes a means to regulate gene expression post-transcriptionally. Aberrant splicing may lead to the production of transcripts that could encode potentially deleterious proteins. However relevant non-coding RNAs may be in disease, these have to the best of our knowledge not been comprehensively examined in BPD brain tissue.

To comprehensively probe the implication of numerous RNA classes in BPD, we performed the first multi-class RNA sequencing experiment in the frontal gyrus of BPD patients, regardless of polyadenylation status. We thus compared the relative abundances of both protein-coding and non-coding RNAs in BPD with healthy controls. We find several differentially expressed genes (DEGs) and long lncRNA transcripts.

## 2. Methods

### 2.1. Study Population

All methods were carried out in accordance with relevant guidelines and regulations. Postmortem material from all BPD and healthy control subjects used in this study was obtained from the Netherlands Brain Bank (NBB), Netherlands Institute for Neuroscience, Amsterdam, the Netherlands (open access: www.brainbank.nl). All experimental protocols were approved by the NBB where this project was assigned id number 909. All material has been collected from donors for or from whom written informed consent was obtained by the NBB. Procedures, information—and consent forms of the NBB have been approved by the Medical Ethics Committee of the VU Medical Centre at 30 April 2009 (https://www.brainbank.nl/media/uploads/file/Ethical%20declaration% 202019.pdf). The informed consent includes permission for a brain autopsy and the use of the material and clinical information for research purposes. A first cohort that consisted of four cases and four controls—matched on gender, age, post-mortem interval (<9 h), and pH (6.0–7.0)—was used for RNA sequencing experiments. A second cohort also consisting of four cases and four controls and matched on age, post-mortem interval (<9 h), and pH (= 6.0–7.0) was used for replication experiments. Extensive phenotype information available from the NBB was used to verify diagnosis and healthy control status. An overview of detailed phenotypic information (including medication) is presented in the Appendix A. In summary, two patients were using lithium and one was using valproic acid in the 24 h prior to death, while all patients had been on either lithium or valproic acid in the months preceding their death. BPD causes of death included: euthanasia (one patient), cardiac arrest (one patient), renal insufficiency (one patient), lung carcinoma (one patient), neck trauma and pneumonia (one patient), and cachexia and dehydration (three patients, one of whom also had possible lithium intoxication). Causes of death for healthy controls were: renal insufficiency (one control), euthanasia (two controls), superior mesenteric artery thrombosis (one control), cachexia due to cancer (one control), pulmonary insufficiency (one control), heart failure (one control), and breast cancer (one control).

### 2.2. Human Tissue and RNA Processing

Frozen medial frontal gyrus tissue from each subject was powdered with the help of a grinder in a pre-chilled mortar to prevent thawing. Total RNA was then extracted from 20–40 mg of the powdered frozen samples using the RNeasy Plus Mini Kit (Qiagen, Hilden, Germany) according to the manufacturer’s instructions and then DNase-treated to eliminate genomic DNA contamination. Finally, RNA integrity (RIN) was verified on an Agilent 2100 bioanalyzer (Agilent Technologies, Santa Clara, CA, USA) and only samples with RIN ≥ 5.0 were used for RNA sequencing. Detailed information about postmortem tissue conditions is provided in the Appendix A.

### 2.3. RNAseq Library Preparation, Sequencing and Alignment

A total of 1 ug of RNA per sample in a 5-μL volume was rRNA depleted using the Ribo-Zero Magnetic Kit (human/mouse/rat; Epicentre, Madison, Wisconsin, USA) before library preparation. Stranded, paired-end sequencing libraries were prepared by the ServiceXS sequencing facility (BaseClear, Leiden, The Netherlands) and sequenced on the Illumina HiSeq 2500 platform (Illumina, San Diego, CA, USA). The reads (2 × 100 bp, mapping to both exons and introns) were de-multiplexed and converted to FASTQ format using CASAVA software from Illumina by the ServiceXS sequencing core facility. FASTQ files were mapped to the hg19/GCh37 reference human genome (iGenomes, San Diego, California, USA) with TopHat2 (version 2.0.13) [42], using the ‘fr-firststrand’ option for strand orientation to generate BAM-formatted genomic coordinates. For lncRNA alignment, reads were aligned using Tophat2 aligner against the NONCODEv4 [43] reference database that contains 54,073 human annotated lncRNA sequences.

### 2.4. Differential Expression (DE) and Weighted Gene Co-expression Network Analyses (WGCNA)

Count data for genes and lncRNA transcripts were analyzed in R (www.r-project.org) using the Bioconductor package edgeR version 3.12.1 [44] with the trimmed mean of M-values (TMM) normalization method [45]. A generalized linear model was used to test the null hypothesis of absence of differential expression between the two groups. Gene expression levels were corrected for gender effect by including sex as a covariate in the model. P-values adjusted for multiple testing were calculated using Benjamini Hochberg false discovery rate (FDR) and only genes at FDR < 0.05 were considered significantly DE. Volcano plots and expression MA scatter plots were generated in R using the ggplot2 library. To establish in which cell types the differentially expressed genes are expressed, we used an online tool that provides gene-based transcriptomics results parsed by cell type [46]. This tool outputs fragments per kilobase of transcript sequence per million mapped fragments (FPKM) in each possible brain cell type (astrocytes, neurons, oligodendrocyte precursor cells (OPCs), newly formed oligodendrocytes, myelinating oligodendrocytes, microglia and endothelial cells). WGCNA analysis methods are described in the Appendix A.

### 2.5. Validation and Replication Analyses

Lower RIN values than 5 were allowed for validation and replication experiments, which is in line with previous findings demonstrating reliable expression data derived from tissues with fairly low RIN values in qt-PCR [47]. No data were available publicly or available in this cohort to account for cell type composition. The unpaired Welch t-test was used to examine case-control differences in gene expression levels in these validation and replication steps, with a 95% confidence cutoff.

Furthermore, to test the concordance between our DEG results and previous RNAseq findings in BPD, using Pearson correlation analyses in R (www.r-project.org) we compared the log2-fold change values of all genes in our dataset with two independent brain tissue RNAseq datasets [13].

### 2.6. Gene Ontology (GO) term and Gene Set Enrichment Analyses (GSEA)

DEGs at FDR < 0.1 were entered into the GO-term R package goseq [48] to correct for bias due to transcript length. Gene set tests were conducted using the fry function [49] that runs an infinite number of rotations to test whether a set of genes is differentially expressed by assessing the entire set of genes as a whole.

### 2.7. Alternative Splicing (AS) Detection

To identify alternative splicing events, we used the Bioconductor package spliceR [50] that uses the output from any full-length RNA-seq assembler to seek for single (ESI) or multiple (MESI) exon skipping, alternative donor and acceptor sites, intron retention (IRI), alternative first (A5) or last exon (A3) usage, and mutually exclusive exon (MEE) events. We ran the spliceR workflow using output from the RNAseq assembler Cufflinks [51] to detect such events. As spliceR does not generate summary statistics we performed a chi-squared test to assess whether the total number of AS events and the numbers of each of these classes differed between cases and controls. We also ran a chi-squared test to examine any possible differences between the number of genes carrying each class of AS events and the total number of genes potentially carrying AS events. We again adjusted the *p*-values for multiple testing by using Benjamini Hochberg FDR correction and only AS events with FDR < 0.05 were considered significant. Enrichment analysis of associated transcripts at FDR < 0.1 was performed with the Bioconductor package GOstats [52] that uses the hypergeometric distribution to test for overrepresentation. To interpret those results reliably, the resulting GO terms and FDR adjusted *p*-values were analyzed with the GO interpretation tool REVIGO [53] to group the terms and reduce redundancy using “medium” as the size of the output list of terms and “Homo sapiens” as reference.

### 2.8. Detection and Expression Analysis of Circular RNAs

Detection of circRNA was performed using the python scripts of the find_circ pipeline [33], which is available from circBase at http://circbase.org/cgi-bin/downloads.cgi [33]. Briefly, after FASTQ alignment using Tophat2, the unmapped reads were used as inputs for the find_circ tool to identify back-spice sites as described by the developers. Since find_circ returns bed files, IDs were assigned by matching the genomic coordinates of every identified circle to those of its host gene by BEDTools [54]. For differential expression, count data were analyzed in R using the edgeR with TMM normalization in the same generalized linear model as explained above for gene and lncRNA differential expression analyses. To reduce the likelihood of type-I errors (although at the cost of increased type-II error risk), we set the FDR significance threshold for this first analysis of circRNAs in BPD at < 0.1.

### 2.9. Code and Data Availability and Data Sharing

Codes used for the statistical analyses and the data that support the findings of this study are available on request from the corresponding author and may then be shared. Upon publication the code will be shared on Github. The data are not publicly available due to privacy or ethical restrictions.

## 3. Results

### 3.1. Baseline Characteristics

Baseline characteristics were similar for BPD patients and controls (Table 1 and Appendix A). The average number of reads was ~63 million with 71% of all reads mapping to the human genome. To rule out that other variables than disease status (e.g., RNA integrity, age, cell type composition, and postmortem interval) may be driving case-control differences, we first carefully matched cases and controls, then performed gene expression principal components analysis (PCA, Appendix A) and finally validation and replication experiments.

### 3.2. Coding RNA: Differential Expression, Network and Enrichment Analyses

At FDR < 0.05, our differential expression analysis revealed 20 significantly differentially expressed genes (DEGs; Appendix A). At FDR < 0.1, we found 36 DEGs, all of which were upregulated in BPD (Figure 1A). RT-PCR and RT-qPCR (Appendix A) were then used to (1) validate our differentially expressed gene (DEG) findings within the discovery cohort; and (2) replicate these findings in an independent replication cohort. To that end, we selected primers for the top three DEGs and two randomly chosen DEGs. We thus successfully validated this entire predefined subset of DEGs (*CD93, SOCS3, BCL6B, PODXL* and *ABCA1*; Figure 1B and Appendix A). We also successfully replicated four out of five of these genes (*CD93, SOCS3, PODXL* and *ABCA1*) in our independent replication cohort (Figure 1C and Appendix A). Although *BCL6B* did not statistically replicate in this cohort, its expression level aligned with the RNAseq data (Figure 1C). In addition, we found a significant correlation in log2-fold change values between all genes in our dataset and both published datasets [13] (r = 0.39 and r = 0.29; both *p*-values < 2.2 × 10^−16^; Appendix A).

We then proceeded to assess cell-type specificity of the top five differentially expressed genes and found all but one of those genes having more than 10-fold higher expression levels in endothelial tissue than any other cell type (Appendix A).

Using weighted gene co-expression network analyses (WGCNA) we were unable to identify any significantly correlated modules. Gene ontology (GO) term enrichment analysis, however, showed significant enrichments of biological processes related to angiogenesis and vascular system development (Appendix A). In our gene-set enrichment analysis (GSEA), we identified enrichment in the GO term GO:0034720 “Histone H3-K4 demethylation” (FDR = 0.00017; Appendix A): the expression levels of genes associated with this GO term were upregulated in BPD compared to controls.

### 3.3. Alternative Splicing (AS) Events

The total number of AS events was higher in BPD subjects than in controls (137,017 in BPD versus 129,294 in controls, p = 1.23 × 10^−50^; Appendix A). Many of these AS events occur in DEGs (Appendix A) and in BPD-susceptibility genes (Appendix A; derived from whole-exome sequencing studies [8,9,10]. It is currently unknown whether DEGs may sometimes be explained by AS events. We also observed significant increases in single (ESI) and multiple (MESI) exon skipping, intron retention (IRI), and alternative first (A5) and last exon (A3) usage in BPD brains of 4.4%, 5.9%, 6.3%, 8.3% and 8.9% (all five adjusted *p*-values ≤ 6.5 × 10^−6^; Appendix A), respectively. The number of mutually exclusive exon (MEE) events was similar for BPD patients and controls. The total number of genes carrying AS events was also increased in BPD patients (adjusted p = 1.3 × 10^−13^; Appendix A).

The functional analysis of BPD-specific AS events indicated significant GO term enrichment for categories associated with cell-cycle, post-translational protein modification, protein transport/localization, RNA splicing and processing for the ESI, MESI and IRI classes; in addition, A5 BPD-specific events were significantly overrepresented in the mitochondrial respiratory chain category (Appendix A).

### 3.4. Differentially Expressed Long Non-Coding RNAs in BPD

We detected expression from 27,348 lncRNA transcripts in both patients and controls. These detected lncRNAs exhibited an overall lower expression than protein-coding genes (t = 42.27; p = 1.08 × 10^−9^; Appendix A). The differential expression analysis identified a total number of ten differentially expressed lncRNAs (six upregulated and four downregulated lncRNAs) in BPD patients at FDR < 0.05, while at FDR < 0.1 these numbers were 11 and four, respectively (Figure 2 and Appendix A).

### 3.5. Circular RNA Transcripts in Healthy and Affected Brain Tissue

We identified a reservoir of 22,530 circRNAs with a minimum of two reads spanning the back-splice junction in the medial frontal gyrus. Of the 22,530 circRNAs identified, 21,019 come from 5906 annotated loci (3532 of which give rise to multiple circles), while 1511 circRNAs map outside the genomic region of known genes (Figure 3A,B; Appendix A). When applying a log fold change cutoff of ≥ 1.5 or ≤ −1.5 irrespective of statistical significance, 689 circRNAs were either upregulated or downregulated in BPD (Appendix A), one of which (*ZDHHC11*) overlapped with the DEGs. Three additional circRNA transcripts encompass this locus (Appendix A). At the 104 BPD risk genes based on the whole-exome sequencing studies, five circRNA transcripts are upregulated at a log fold change cutoff of ≥ 1.5 and 1 is downregulated at ≤ −1.5 (Appendix A).

Differential expression analysis of the predicted circRNA transcripts between patients and controls identified one circRNA at FDR < 0.05 (*cNEBL*) and one additional circRNA at FDR < 0.1 (*cEPHA3*; Figure 3C). As we did not find significant differences in expression of linear *NEBL* and *cEPHA3* (Appendix A), the differential expression of these transcripts between cases and controls is unlikely to be driven by their linear counterparts.

## 4. Discussion

We here demonstrate differences in both coding and non-coding RNA segments between bipolar disorder (BPD) and healthy controls. By applying the bioinformatics pipeline that we developed to total RNA libraries we first signal several differentially expressed genes and then validate and replicate our findings by PCR. We then demonstrate that alternative splicing events are increased in BPD and that expression levels of several lncRNAs differ between cases and controls. Finally, we describe a reservoir of circular RNAs in human medial frontal cortex and altered levels of circular RNA transcripts in BPD.

Functional annotation analysis of the differentially expressed genes (DEGs) showed significant enrichments of biological processes related to angiogenesis and vascular system development. The largest BPD GWAS did not find evidence of angiogenesis or vascular system development being implicated in BPD [7], which may be related to either statistical power (that was increased in the GWAS) or differences between CNS processes probed by RNAseq of brain tissue and whole-blood derived DNA that is used for GWAS. To assess whether the DEGs described here overlap with the largest BPD whole-genome RNA study, we performed a look-up of our 36 DEGs in Gandal et al. [12]. We thus found evidence of involvement of *STAB1* in that study too (at p = 0.0013 and FDR-corrected p = 0.039 in that study). The other 35 DEGs highlighted by us were not found to be associated with BPD in that study [12], which may be due to statistical power or diverging brain regions studied. Similarly, we did not detect the two lncRNAs reported in the lithium GWAS [55]. The results of our cell-type specificity analyses pointing to endothelial tissue agree with vascular tissue being implicated in bipolar disorder. This alignment between pathway and cell-type analyses is intriguing since, to our knowledge, no other study had reported a role for genes involved in angiogenesis and vascular development in BPD. Meta-analyses of fMRI studies in BPD point to frontal hypoactivation [56,57]. Based on our findings, one may postulate that impaired angiogenesis may be at the root of such hypoactive areas in the BPD brain; on the other hand, increased angiogenesis may constitute a compensatory mechanism for such hypoactivity. Whether microvascular changes have a role in the *neurobiology* of BPD remains to be unraveled. An alternative explanation for this pathway finding relates to medication use in BPD as lithium and valproic acid promote angiogenesis [58,59], even in infarcted brain areas [60]. A potential mechanism at play here may be that these agents increase expression levels of genes involved with angiogenesis. Since all study participants had at death or previously used one of these two mood stabilizers (Appendix A), our results may reflect a consequence of treatment rather than disease mechanisms. Future studies aimed at collecting postmortem brain tissue of medication-naïve BPD patients would be highly laborious and time-consuming, yet not unconceivable.

Using gene-set enrichment analysis (GSEA) we identified enrichment in histone H3-K4 demethylation: the expression levels of genes associated with this GO term were upregulated in BPD brains compared to controls. Numerous studies indicate that histone modifications have important roles in BPD [61], although the most recent GWAS did not implicate this pathway [7]. One of the best characterized histone modifications is H3-K4 methylation, an epigenetic phenomenon highly enriched at transcriptional start sites (TSS) and associated with active transcription. The design of the current study did not allow us to investigate H3-K4 methylation status on the TSS of genes in BPD brains. Future studies integrating gene expression with methylation probes within the same cohort may enable such analyses.

CircRNAs are increasingly being discovered as a functionally important kind of ubiquitous, endogenous non-coding RNAs [33,34]. Recent evidence reveals that circRNAs function as miRNA sponges [32] and regulate parent gene expression, predisposing subjects to certain diseases [33,34]. Despite the importance of circRNAs to several types of cancer [62], the contribution of this highly stable class of RNAs to (neuro)psychiatric disorders, including BPD, had been explored in only few studies. Here, we reveal two circRNAs to be upregulated in BPD. These circRNAs derive from the *NEBL* and *EPHA3* loci, respectively. Particularly *EPHA3* may be of relevance to BPD as it is implicated in developmental events, mostly in the central nervous system (CNS). The Eph receptor belongs to the ephrin receptor subfamily of the protein-tyrosine kinase family. Eph receptors and ephrins (their ligands) regulate key CNS processes, such as neurotransmitter release, postsynaptic glutamate receptor conductance, synaptic glutamate reuptake, and dendritic spine morphogenesis [63]. They are involved in memory formation [64] and anxiety [65]. Memory deficits (during episodes and interepisodic) and high anxiety levels (during episodes) are both often reported in BPD. It is therefore of interest that Ephs and ephrins (their ligands) may be pharmacologically targeted [63,66,67]: EphB2-blockade impedes physiological stress responses in wild-type mice [65]. In addition to serving as therapeutic targets, traces of RNAs may serve as markers to monitor disease progression, as has been demonstrated for cancer [62]. Although one previous whole-transcriptome study on BPD brain tissue also used an rRNA-depleted library, circRNAs and AS events were not reported, while lncRNA results did not exceed significance thresholds [15]. If future studies confirm our BPD circRNA findings in peripheral tissues, circRNA molecules could one day aid in the diagnostic workup of patients suspected of a bipolar spectrum disorder.

Previous RNAseq studies in BPD have focused on other brain regions, in particular hippocampus and dorsal striatum tissue where immune response pathways were found to be implicated [16,17]. A different study on hippocampus tissue detected aberrant miR-182 signaling [14]. Other authors deep sequenced dorsolateral prefrontal cortex and detected dysregulation of neuroplasticity, circadian rhythms and GTPase binding in BPD [13]. Finally, anterior cingulate sequencing revealed G protein-receptor dysregulation [15]. The current findings thus add to a quite heterogeneous array of biological pathways implicated in BPD based on whole transcriptomics studies, which in part may be owing to the disparate brain regions targeted in these studies. Future studies incorporating extremely large sample sizes while comparing coding and non-coding RNAs in all possible brain regions of BPD patients with healthy controls may disentangle the state and region-specificity of such pathways to the disease.

Several limitations should be borne in mind when interpreting the results of the current study. First, the modest sample size may have impeded power to detect any significant modules in our WGCNA. Second, having chosen a non-polyadenylated library as opposed to a small RNA library by consequence precluded us from examining miRNAs, molecules of particular relevance to schizophrenia. Third, having focused on coding and non-coding RNA in rather unique and scant sections of postmortem tissue prevented us from designing analyses at the protein level, a limitation that future studies may be able to overcome. Fourth, we chose to target a single brain area which comes with the disadvantage of lack of generalizability of our findings to other brain regions. And finally, while our circRNA findings in both healthy and affected brain tissues are intriguing, this relatively unexplored domain of RNA molecules currently does not allow cross-validation in other datasets. These results consequently will have to await confirmation in other cohorts. Such cohorts will need to be better powered than the current study as power for replication in our modestly sized cohort is limited in particular for non-coding RNAs. Moreover, future studies may disentangle the specificity of our findings to BPD by incorporating other psychiatric disorders.

In conclusion, our study hints that RNA dysregulation in bipolar disorder is not limited to coding regions.

## Figures and Tables

**Figure 1 genes-10-00946-f001:**
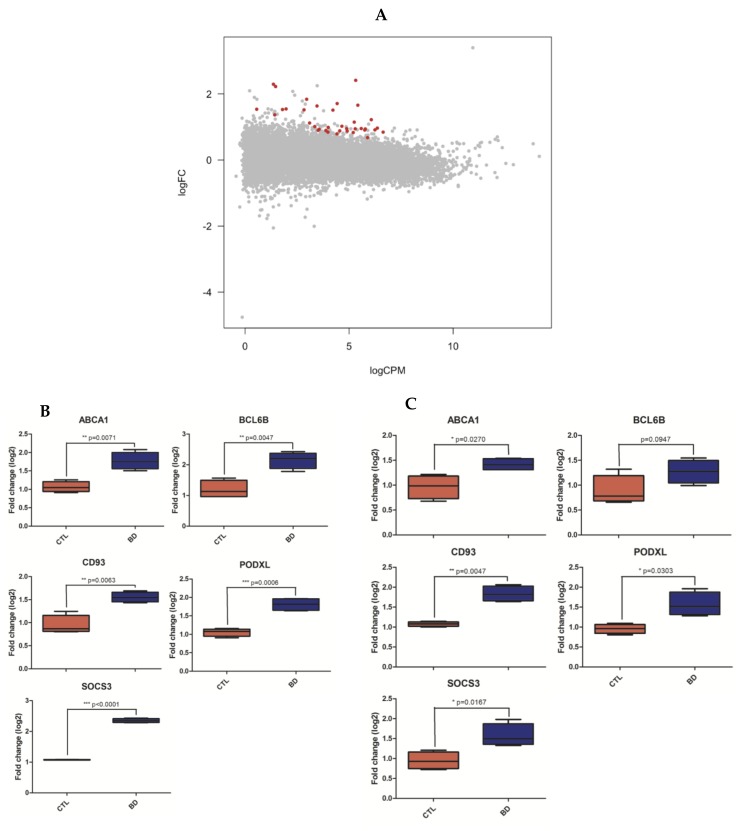
Differentially expressed genes analyses. Plot showing the 36 significant differentially expressed genes (DEGs) at FDR < 0.1 in subjects with BPD (blue) versus healthy controls (red, **A**). Validation (**B**) and replication (**C**) of RNAseq gene expression analysis by RT-qPCR. RNA level differences were calculated by unpaired Welch t-test (* p < 0.05, ** p < 0.01, *** p < 0.005). All 5 DEGs were successfully validated by RT-PCR and had significant increased expression levels in BPD patients compared to controls (**B**). Four out of five DEGs were successfully replicated and had increased expression levels in BPD patients compared to controls; only BCL6B did not replicate significantly, although its expression level aligned with the sequencing data (**C**).

**Figure 2 genes-10-00946-f002:**
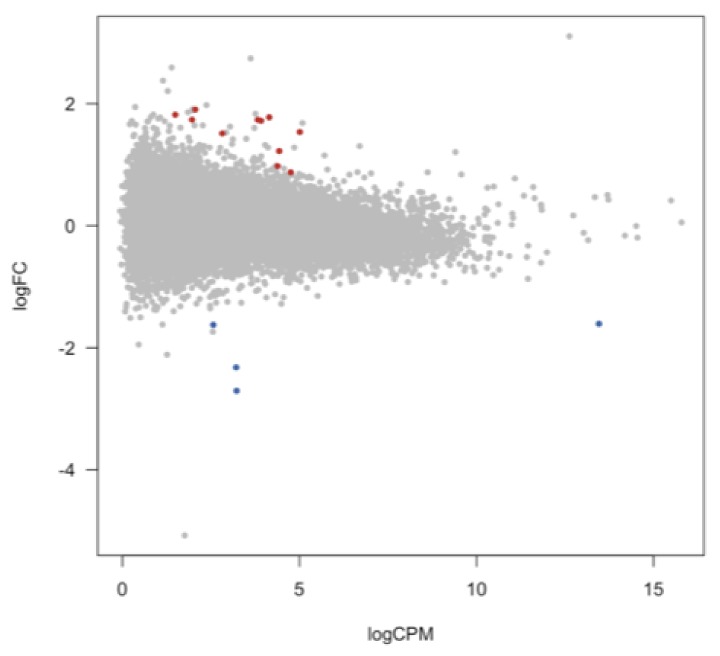
Differential regulation of lncRNAs in medial frontal gyrus in bipolar disorder. This MA plot depicts the 15 differentially expressed lncRNA transcripts at FDR < 0.1. On the *x*-axis, the logCPM is the log-transformed average expression level (expressed in CPM, counts per million mapped reads) for each gene across the two groups; on the *y*-axis, the logFC represents the log of the ratio of expression levels for each gene between two experimental groups. In red are the upregulated transcripts; in blue the downregulated transcripts.

**Figure 3 genes-10-00946-f003:**
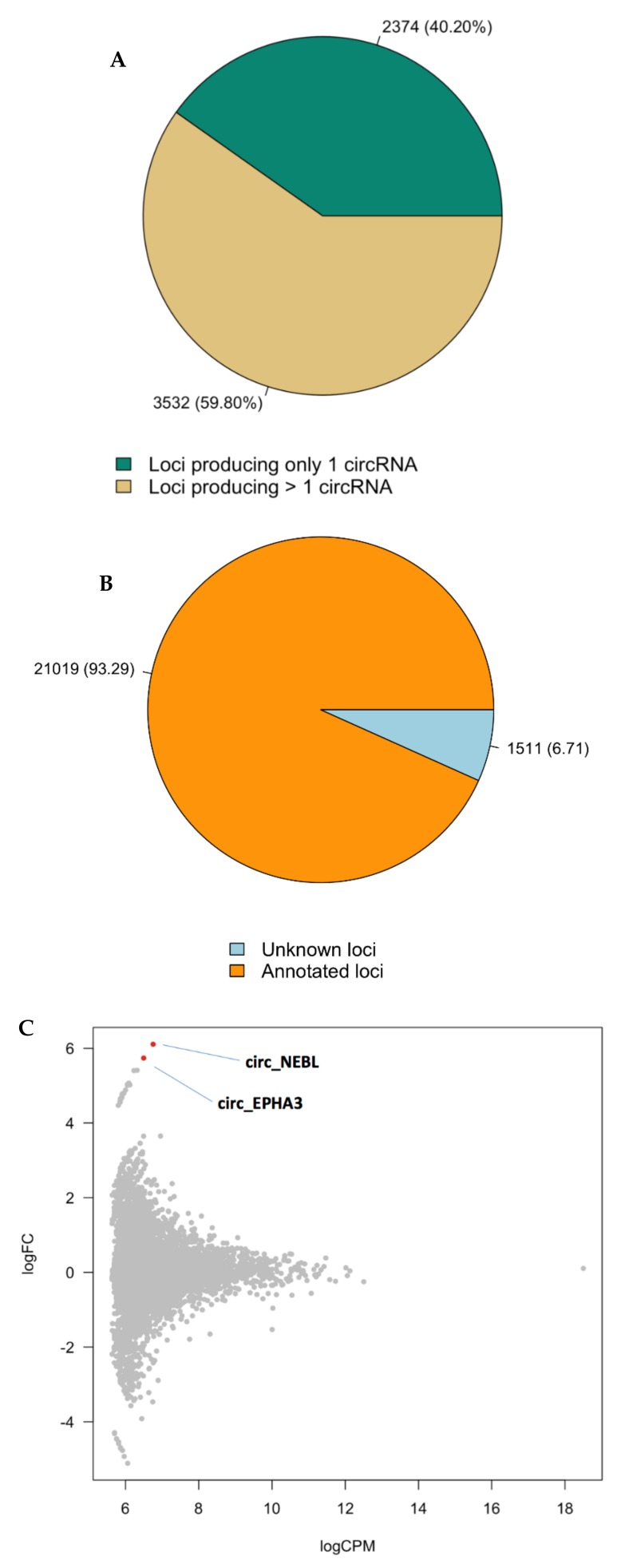
Differential expression of circRNAs in BPD brains. Pie-chart representing the numbers and the percentages of the circRNAs from annotated loci and those from unknown loci, respectively (**A**). Pie-chart representing the numbers and percentages of loci producing only one circRNA and those giving rise to multiple circles, respectively (**B**). MA plot showing the two differentially expressed circular RNAs at FDR < 0.1 in subjects with BPD versus healthy controls (**C**).

**Table 1 genes-10-00946-t001:** Baseline characteristics of the study population. PMI = postmortem interval. M = male. F = female.

Cohort	Discovery (*N* = 8)		Replication (*N* = 8)	
	Controls (*N* = 4)	Bipolar Disorder (*N* = 4)	Controls (*N* = 4)	Bipolar Disorder (*N* = 4)
PMI	6 h 55	6h 25	6 h 47	5 h 21
PMI range	6 h 15–7 h 15	4 h 50–8 h 00	5 h 05–8 h 10	4 h 35–6 h 40
pH	6.41	6.39	6.53	6.48
pH range	6.23–6.76	6.26–6.53	6.47–6.58	6.38–6.70
Sex	1 F, 3 M	1 F, 3 M	3 F, 1 M	4 M
Mean age	65.7	72.5	70	73.5
Age range	49–88	70–79	57–93	68–81

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
