# Peer review of "Coding and Non-Coding RNA Abnormalities in Bipolar Disorder"

_genes, 2019, doi:10.3390/genes10110946_

Round 1

Reviewer 1 Report

The authors describe a post-mortem brain medial frontal gyrus gene expression study of BPD. The paper is straightforward and clearly presented.

Points to consider:

In the Introduction the authors state (p 2), “Such genome-wide studies have not provided the biological insight as has been the case for schizophrenia.” The authors refer to two BPD GWAS reports, published 8 and 3 years ago. They should refer to Stahl et al, Nature Genetics, 2019, who, describe > 30 genome-wide significant loci, including multiple loci shared with schizophrenia, and they should delete the quoted sentence. The authors should specify the ethnicity, medications taken chronically during the several months prior to death and cause of death for each case and control. The annotation analyses, specifying angiogenesis and vascular system development stands in contrast to the pathways identified by Stahl et al (2019). Can the authors comment? The most significant limitation of the study is the small number of patient and control post-mortem brain samples studied. A second limitation is the single brain area selected for analysis. Several years ago, a GWAS of lithium response in BPD was published (Lancet 387:1085, 2016). These authors noted two lnc RNAs (AL157359.3 and AL157359.4) were genome-wide significant. Were these detected in medial frontal gyrus brain region, and what were the results? Were the pathways detected also found by Stahl et al (2019)?

Author Response

The authors describe a post-mortem brain medial frontal gyrus gene expression study of BPD. The paper is straightforward and clearly presented.

> Points to consider:

In the Introduction the authors state (p 2), “Such genome-wide studies have not provided the biological insight as has been the case for schizophrenia.” The authors refer to two BPD GWAS reports, published 8 and 3 years ago. They should refer to Stahl et al, Nature Genetics, 2019, who, describe > 30 genome-wide significant loci, including multiple loci shared with schizophrenia, and they should delete the quoted sentence.

We thank the reviewer for pointing our omission out. We have inserted the reference and have deleted the sentence suggested by the reviewer. Instead, we have included a sentence that reads: ‘The best powered GWAS has highlighted 30 loci for BPD and has provided insight into genes and pathways involved in the disease [7].

The authors should specify the ethnicity, medications taken chronically during the several months prior to death and cause of death for each case and control.

We agree the information given about the participants was scant. We have therefore added the requested information with the exception of ethnicity that is unavailable, in the following sentences added to the methods: ‘In summary, two patients were using lithium and one was using valproic acid in the 24 hours prior to death, while all patients had been on either lithium or valproic acid in the months preceding their death. BPD causes of death included: euthanasia (1 patient), cardiac arrest (1 patient), renal insufficiency (1 patient), lung carcinoma (1 patient), neck trauma and pneumonia (1 patient), and cachexia and  dehydration (3 patients, one of whom also had possible lithium intoxication). Causes of death for healthy controls were: renal insufficiency (1 control), euthanasia (2 controls), superior mesenteric artery thrombosis (1 control), cachexia due to cancer (1 control), pulmonary insufficiency (1 control), heart failure (1 control), and breast cancer (1 control).

The annotation analyses, specifying angiogenesis and vascular system development stands in contrast to the pathways identified by Stahl et al (2019). Can the authors comment?

We indeed had not sufficiently emphasized the lack of agreement between our finding and the 2019 Stahl GWAS. We now do so in a sentence that reads: ‘The largest BPD GWAS did not find evidence of angiogenesis being implicated in BPD  [7], which may be related to either statistical power (that was increased in the GWAS) or  differences between CNS processes probed by RNAseq of brain tissue and whole-blood derived DNA that is used for GWAS.

The most significant limitation of the study is the small number of patient and control post-mortem brain samples studied. A second limitation is the single brain area selected for analysis.

We indeed had discussed the sample size as a limitation but had not mentioned the single brain region as a limitation. This additional limitation is discussed in the new version as the fourth limitation now: ‘Fourth, we chose to target a single brain area which comes with the disadvantage of lack of generalizability of our findings to other brain regions.

Several years ago, a GWAS of lithium response in BPD was published (Lancet 387:1085, 2016). These authors noted two lnc RNAs (AL157359.3 and AL157359.4) were genome-wide significant. Were these detected in medial frontal gyrus brain region, and what were the results? Were the pathways detected also found by Stahl et al (2019)?

We had indeed omitted a comparison between the pathways that we detected and those reported in Stahl et al. We now mention in the discussion in two sections that angiogenesis and vascular system development, as well as histone demethylation, were not found in that previous GWAS. We also now mention we did not find these two lncRNAs of the lithium response GWAS.

The largest BPD GWAS did not find evidence of angiogenesis or vascular system development being implicated in BPD  [7], which may be related to either statistical power (that was increased in the GWAS) or differences between CNS processes probed by RNAseq of brain tissue and whole-blood derived DNA that is used for GWAS. Similarly, we did not detect the two lncRNAs reported in the lithium GWAS [55].

And

“although the most recent GWAS did not implicate this pathway [7]. ‘’

Reviewer 2 Report

This study uses whole transcriptome sequencing to identify genes that are differentially expressed in the medial frontal gyrus from patients with bipolar disorder. Although the number of subjects is very small (4 cases and 4 controls for sequencing, 4 cases and 4 controls replication), a novel and interesting aspect of this study is that the authors explore non-coding RNA species such as circRNAs, reporting some to be differentially expressed. The methods are essentially sound and the manuscript well-written. I have the following suggestions for improvement of the ms:

The authors state in the introduction that GWAS for bipolar disorder have not provided the insights that schizophrenia GWAS have. However, they cite early GWAS for bipolar disorder that had much less statistical power. They should revise this section citing the most recent bipolar disorder GWAS (Stahl et al, Nature Genetics 2019), where 30 genome-wide significant loci are identified. The authors should state whether any of their differentially expressed genes are also DE in the DLPFC of individuals with bipolar disorder in the much larger study of Gandal et al (Science 2018). In the Discussion, the authors state that functional annotation of DE genes might not have picked up the same terms if polyadenylated RNA-Seq libraries had been used. However, most GO annotations are based on protein-coding genes which have polyA tails, so this isn’t likely. Page 11, line 358: ‘disparaging brain regions’ should probably be ‘disparate’. In the Discussion, the authors state that the ratio of DE genes to all RNAs of that class was highest for protein coding genes, then lncRNA, then circRNA. They state that they believe that these differences cannot be accounted for by technical characteristics. However, protein coding genes are usually expressed at a higher level than non-coding RNA, which will result in better estimations of expression by RNA-Seq and therefore a bias towards more DE genes detected, so I think this section should be deleted.

Author Response

Reviewer 2:

This study uses whole transcriptome sequencing to identify genes that are differentially expressed in the medial frontal gyrus from patients with bipolar disorder. Although the number of subjects is very small (4 cases and 4 controls for sequencing, 4 cases and 4 controls replication), a novel and interesting aspect of this study is that the authors explore non-coding RNA species such as circRNAs, reporting some to be differentially expressed. The methods are essentially sound and the manuscript well-written. I have the following suggestions for improvement of the ms:

The authors state in the introduction that GWAS for bipolar disorder have not provided the insights that schizophrenia GWAS have. However, they cite early GWAS for bipolar disorder that had much less statistical power. They should revise this section citing the most recent bipolar disorder GWAS (Stahl et al, Nature Genetics 2019), where 30 genome-wide significant loci are identified.

We thank the reviewer for pointing our omission out. We have inserted the reference and have deleted the sentence suggested by the reviewer. Instead, we have included a sentence that reads: ‘The best powered GWAS has highlighted 30 loci for BPD and has provided insight into genes and pathways involved in the disease [7].

The authors should state whether any of their differentially expressed genes are also DE in the DLPFC of individuals with bipolar disorder in the much larger study of Gandal et al (Science 2018).

We had indeed not well discussed this paper and the degree of overlap with our study. We now refer to this reference in the introduction and discussion. As per the reviewer’s suggestion we also performed a look-up of DEGs in Gandal et al and found one DEG described by us overlapping with Gandal et al. We added this to the discussion section: “To assess whether the DEGs described here overlap with the largest BPD whole-genome RNA study, we performed a look-up of our 36 DEGs in Gandal et al [12]. We thus found evidence of involvement of STAB1 in that study too (at p=0.0013 and FDR-corrected p=0.039 in that study). The other 35 DEGs highlighted by us were not found to be associated with BPD in that study [12], which may be due to statistical power or diverging brain regions studied.

In the Discussion, the authors state that functional annotation of DE genes might not have picked up the same terms if polyadenylated RNA-Seq libraries had been used. However, most GO annotations are based on protein-coding genes which have polyA tails, so this isn’t likely.

We agree and have deleted the sentence.

Page 11, line 358: ‘disparaging brain regions’ should probably be ‘disparate’.

We agree and have changed this accordingly.

In the Discussion, the authors state that the ratio of DE genes to all RNAs of that class was highest for protein coding genes, then lncRNA, then circRNA. They state that they believe that these differences cannot be accounted for by technical characteristics. However, protein coding genes are usually expressed at a higher level than non-coding RNA, which will result in better estimations of expression by RNA-Seq and therefore a bias towards more DE genes detected, so I think this section should be deleted.

We agree and have deleted the entire paragraph as none of it really does contribute to the discussion.

Round 2

Reviewer 1 Report

The authors have adequately responded to each of the points below.

" Points to consider:

In the Introduction the authors state (p 2), "Such genome-wide studies have not provided the biological insight as has been the case for schizophrenia." The authors refer to two BPD GWAS reports, published 8 and 3 years ago. They should refer to Stahl et al, Nature Genetics, 2019, who, describe > 30 genome-wide significant loci, including multiple loci shared with schizophrenia, and they should delete the quoted sentence. The authors should specify the ethnicity, medications taken chronically during the several months prior to death and cause of death for each case and control. The annotation analyses, specifying angiogenesis and vascular system development stands in contrast to the pathways identified by Stahl et al (2019). Can the authors comment? The most significant limitation of the study is the small number of patient and control post-mortem brain samples studied. A second limitation is the single brain area selected for analysis. Several years ago, a GWAS of lithium response in BPD was published (Lancet 387:1085, 2016). These authors noted two lnc RNAs (AL157359.3 and AL157359.4) were genome-wide significant. Were these detected in medial frontal gyrus brain region, and what were the results?

7. Were the pathways detected also found by Stahl et al (2019)? "